# SMAD4 promotes somatic-germline contact during murine oocyte growth

**Sofia Granados-Aparici[1,2,3]\*, Qin Yang[1], Hugh J Clarke[1,4]\***

[1]Research Institute, McGill University Health Centre, Montreal, Canada; [2]Present address: Cancer CIBER (CIBERONC), Madrid, Spain; [3]Present address: Pathology Department, Medical School, University of Valencia-INCLIVA, Valencia, Spain; [4]Departments of Obstetrics and Gynecology and Biology, Division of Experimental Medicine, McGill University, Montréal, Canada

**Abstract** Development of the mammalian oocyte requires physical contact with the surrounding granulosa cells of the follicle, which provide it with essential nutrients and regulatory signals. This contact is achieved through specialized filopodia, termed transzonal projections (TZPs), that extend from the granulosa cells to the oocyte surface. Transforming growth factor (TGFβ) family ligands produced by the oocyte increase the number of TZPs, but how they do so is unknown. Using an inducible Cre recombinase strategy together with expression of green fluorescent protein to verify Cre activity in individual cells, we examined the effect of depleting the canonical TGFβ mediator, SMAD4, in mouse granulosa cells. We observed a 20–50% decrease in the total number of TZPs in SMAD4-depleted granulosa cell-oocyte complexes, and a 50% decrease in the number of newly generated TZPs when the granulosa cells were reaggregated with wild-type oocytes. Three-dimensional image analysis revealed that TZPs of SMAD4-depleted cells were longer than controls and more frequently oriented towards the oocyte. Strikingly, the transmembrane proteins, N-cadherin and Notch2, were reduced by 50% in SMAD4-depleted cells. SMAD4 may thus modulate a network of cell adhesion proteins that stabilize the attachment of TZPs to the oocyte, thereby amplifying signalling between the two cell types.

**\*For correspondence:**
sograap@gmail.com (SG-A);
hugh.clarke@mcgill.ca (HJC)

**Competing interest:** The authors declare that no competing interests exist.

## eLife assessment

This study reports an **important** mechanism through which the TGF-beta signaling pathway promotes contacts between oocytes and their surrounding somatic cells by regulating the number of transzonal projections (TZPs) in mice. **Convincing** data support the conclusions. The work will be of interest to biomedical researchers who work on ovarian biology and female fertility.

## Introduction

The development of a healthy oocyte within the ovarian follicle requires continuing contact between the germ cell and the somatic granulosa cells that surround it (*El-Hayek and Clarke, 2016*; *Clarke, 2018*; *Richani et al., 2021*; *Robert, 2021*; *Li and Albertini, 2013*; *Doherty et al., 2022*; *Xie et al., 2023*; *Marchais et al., 2022*). Signals from the granulosa cells initiate the increase in the volume of the oocyte when primordial (non-growing) follicles are activated to enter the growth phase (*Zhang et al., 2014*; *Saatcioglu et al., 2016*; *Nagamatsu et al., 2019*). Subsequently, the granulosa cells provide nucleotides, amino acids, and pyruvate to the growing oocyte via gap junctions that connect the two cell types (*Xie et al., 2023*; *Clarke, 2018*; *Marchais et al., 2022*). When gap junctional communication between the two cell types is genetically disrupted, oocytes do not develop normally and cannot be fertilized (*Gittens and Kidder, 2005*; *Simon et al., 1997*; *Winterhager and Kidder, 2015*;

*Carabatsos et al., 2000*). Once the oocyte reaches full size, it acquires the ability to undergo the final stage of development, termed meiotic maturation, but is prevented from precociously doing so by cyclic GMP that is produced and transferred to it by the granulosa cells, also via gap junctions (*Jaffe and Egbert, 2017*; *Wigglesworth et al., 2013*; *Norris et al., 2009*; *Zhang et al., 2010*). In vitro, oocytes can develop normally when enclosed by granulosa cells but not when the granulosa cells are removed and provided in co-culture, further emphasizing the crucial importance of physical contact between the two cell types (*O'Brien et al., 2003*; *Eppig, 1979*; *Herta et al., 2021*).

In primordial follicles, a single layer of squamous granulosa cells surrounds the oocyte and contact it via adherens junctions (*Mora et al., 2012*; *Jorgensen, 2013*). Following activation of growth, a glycoprotein-rich coat termed the zona pellucida becomes established around the oocyte (*Litscher and Wassarman, 2020*; *Wassarman and Litscher, 2018*). The zona pellucida steadily thickens as growth continues, reaching a final thickness of about 7 μm in mice and 15 μm in humans. Despite the barrier imposed by the zona pellucida, the granulosa cells and oocyte maintain direct contact by a dense network of thin filopodia-like projections, termed transzonal projections (TZPs), that grow out from the granulosa cells through the zona pellucida (*Clarke, 2022*; *Macaulay et al., 2014*; *Baena and Terasaki, 2019*; *Motta et al., 1994*; *Zhang et al., 2021*; *Albertini and Rider, 1994*). The tips of the TZPs establish contact with the oocyte plasma membrane through heterotypic adherens junctions comprising N-cadherin (granulosa cell) and E-cadherin (oocyte) and harbour the gap junctions that link the two cell types (*Mora et al., 2012*) and likely also through focal adhesions (*McGinnis and Kinsey, 2015*). New TZPs are generated throughout follicular growth, thereby increasing intercellular molecular trafficking (*El-Hayek et al., 2018a*). A genetically imposed decrease in the number of TZPs leads to defects in oocyte gene expression and to abnormal meiotic maturation and embryonic development (*Crozet et al., 2023*). Despite the essential role of TZPs during oocyte development in mammals, as well as the presence of analogous structures in non-mammalian species (*Kessel et al., 1985*; *Schroeder, 1981*), the molecular mechanisms that govern their generation and stability are not yet understood.

The TGFβ signalling pathway regulates the functions and interactions of a wide range of cell types under various physiological and pathological conditions (*Derynck and Budi, 2019*; *Peng et al., 2022*). Binding of the ligand to a complex of type I and type II receptors located at the cell surface activates signalling within the cell, most commonly through the members of the SMAD family of transcription factors, although SMAD-independent signalling can also occur (*Aashaq et al., 2022*; *Derynck and Budi, 2019*). Upon phosphorylation of the tetrameric type II and type I receptors and subsequent phosphorylation of receptor-activated SMAD proteins (R-SMADs; SMAD1, 2, 3, 5, and 8), a heterotrimeric complex with SMAD4 is formed which enters the nucleus and regulates gene expression by binding to specific SMAD-binding DNA elements. Although R-SMADs can signal in the absence of SMAD4, evidence indicates that the presence of SMAD4 in the heterotrimer amplifies TGFβ cell signalling by stabilizing the complex and by favouring the binding of other transcriptional co-repressors or co-activators of target genes (*Aragón et al., 2019*; *Guglielmi et al., 2021*).

Genetic, pharmacological, and in vitro studies have revealed multiple roles for TGFβ/SMAD signaling during follicular growth and development (*Patton et al., 2021*; *Pangas and Rajkovic, 2015*). TGFβ signalling via SMAD3 in granulosa cells acts initially in primordial follicles to maintain their quiescence (*Granados-Aparici et al., 2019*; *Kang et al., 2014*) and later, following their activation, becomes excluded from the nucleus in association with rapid granulosa cell growth and proliferation (*Hardy et al., 2018*). The TGFβ superfamily members, growth-differentiation factor (GDF) 9, which signals through SMAD2/3, and bone morphogenetic protein (BMP) 15, which signals through SMAD1/5/8, are produced by the oocyte (*Dong et al., 1996*; *Yan et al., 2001*). Female mice lacking GDF9 rarely generate more than a single layer of granulosa cells (*Elvin et al., 1999b*; *Elvin et al., 1999a*; *Dong et al., 1996*). This defect can be rescued by preventing the production of inhibin, and the doubly mutant follicles can develop at least to the pre-antral stage (*Wu et al., 2004*). Genetic inactivation of SMAD4 in the granulosa cells from growing follicles leads to a reduction in the number of antral follicles and ovulated eggs, as well as the appearance of luteinized follicles containing entrapped oocytes (*Pangas et al., 2006*). Similar phenotypes are observed when both SMAD2 and SMAD3 are inactivated in the granulosa cells (*Li et al., 2008*). These effects are likely due at least in part to a failure of the cumulus layer to expand in response to the ovulatory surge of LH (*Pangas et al., 2006*; *Li et al., 2008*), as expansion is known to require SMAD signalling and to be required

for ovulation (*Dragovic et al., 2007*; *Hao et al., 2022*; *Mottershead et al., 2015*; *Peng et al., 2013*; *Yu et al., 2013*).

Two lines of evidence point to a potential role for SMAD signalling in the generation or maintenance of TZPs. First, deletion of GDF9 leads to both a decrease in the number of TZPs and a failure of a proportion of the TZPs to correctly orient towards the oocyte (*Carabatsos et al., 1998*). Second, TZP number is reduced in follicles grown in vitro following RNAi-mediated depletion of GDF9 in the oocyte (*El-Hayek et al., 2018a*). To test the role of SMAD4 specifically, we used a Cre-lox strategy to inactivate *Smad4* in granulosa cells, as previously described (*Jorgez et al., 2004*; *Pangas et al., 2006*). Using image analysis tools and three-dimensional reconstructions, we show that depletion of SMAD4 in the granulosa cells decreases the number of TZPs that project from individual granulosa cells in both intact granulosa cell-oocyte complexes (GOCs) and in reaggregated complexes. Morphometric parameters including length and orientation are also altered. The decrease in TZP number in intact GOCs is associated with a reduction in the amount of N-cadherin and Notch2 in the granulosa cells, suggesting that the composition of the granulosa cell surface is perturbed in the absence of SMAD4. We propose that SMAD4, through its activity to increase expression of target genes, stabilizes TZPs by promoting their adhesion to the oocyte surface, thereby helping to increase the number of TZPs and communication between the growing oocyte and its follicular microenvironment.

## Results

We began by verifying the expression of SMAD4 during folliculogenesis. As shown in *Figure 1A* and consistent with prior reports, SMAD4 was detectable in primordial and growing follicles as well as in the granulosa cells collected from pre-antral follicles. SMAD2 and SMAD3 were also present, with SMAD2 being the major species detected in oocytes. Two strategies were tested to deplete SMAD4 from the granulosa cells of growing follicles. First, mice were generated carrying two floxed *Smad4* alleles and a transgene encoding Cre recombinase under the control of the *Amhr2* promoter (*Jorgez et al., 2004*; *Pangas et al., 2006*; *Figure 1B*). These mice also carried a transgene (*mTmG*, see Methods) constructed such that Tomato fluorescent protein is expressed in the absence of Cre activity, whereas Cre-mediated recombination switches expression to green fluorescent protein (GFP), thus allowing us to identify cells where Cre had been active. Upon immunostaining of ovarian sections or whole-mount intact follicles, only a small number of cells expressed GFP (*Figure 1C*, left). Consistent with this observation, less than 10% of FACS-sorted granulosa cells were GFP-positive (*Figure 1C*, right). Thus, in our system, *Amhr2-Cre* did not mediate efficient recombination of floxed DNA in granulosa cells. This is consistent with results described in previous reports (*Pangas et al., 2006*; *Li et al., 2008*).

We, therefore, turned to mice that expressed Cre under the control of a tamoxifen-inducible form of the estrogen receptor (ER) gene promoter (*Rosa26^CreERT2*, hereafter termed *ER-Cre*). Because ER is expressed in pituitary gonadotropes and SMAD4 regulates the transcription of the *Fshb* gene encoding the β-subunit of follicle-stimulating hormone (*Li et al., 2018*), this model could not be used in vivo. GOCs from mid-growth stage follicles (diameter 100–120 µm) were, therefore, isolated, exposed to tamoxifen for one day and then incubated in tamoxifen-free medium for another five days. Under these conditions, a substantial fraction of the granulosa cells underwent Cre-mediated recombination, as indicated by the expression of GFP in immunostained GOCs and by FACS analysis (*Figure 1E*). This in vitro system was used for the experiments described here.

To examine the efficiency of Cre-mediated *Smad4* recombination, the mRNA level was assessed in intact GOCs using the RNAscope technique and in purified granulosa cells using qRT-PCR. Following tamoxifen treatment and culture, there was an approximately 60% reduction in the number of foci in the granulosa cells from 5.78+/-0.36 (mean, SEM, n=21) in *ER-Cre^-; Smad4^fl/fl* mice to 2.16+/-0.26 (n=26) in *ER-Cre^+; Smad4^fl/fl* mice (p<0.0001), and a 70% reduction in the amount of PCR product from granulosa cells (n=5, p=0.0001) (*Figure 1F*). Immunoblotting of granulosa cells revealed a corresponding 70% reduction in SMAD4 protein in the granulosa cells of *ER-Cre^+; Smad4^fl/fl* mice compared to *ER-Cre^-; Smad4^fl/fl* controls (n=11, p<0.0001) (*Figure 1G*). Thus, the in vitro ER-Cre strategy substantially reduced although did not eliminate SMAD4 from the granulosa cells.

To examine the effect of SMAD4 depletion on the TZPs, GOCs were isolated and cultured as above, fixed and stained using anti-GFP to label cells where Cre had been active and phalloidin to label the TZPs, and imaged them using confocal microscopy. First, the TZPs were analyzed as a population.

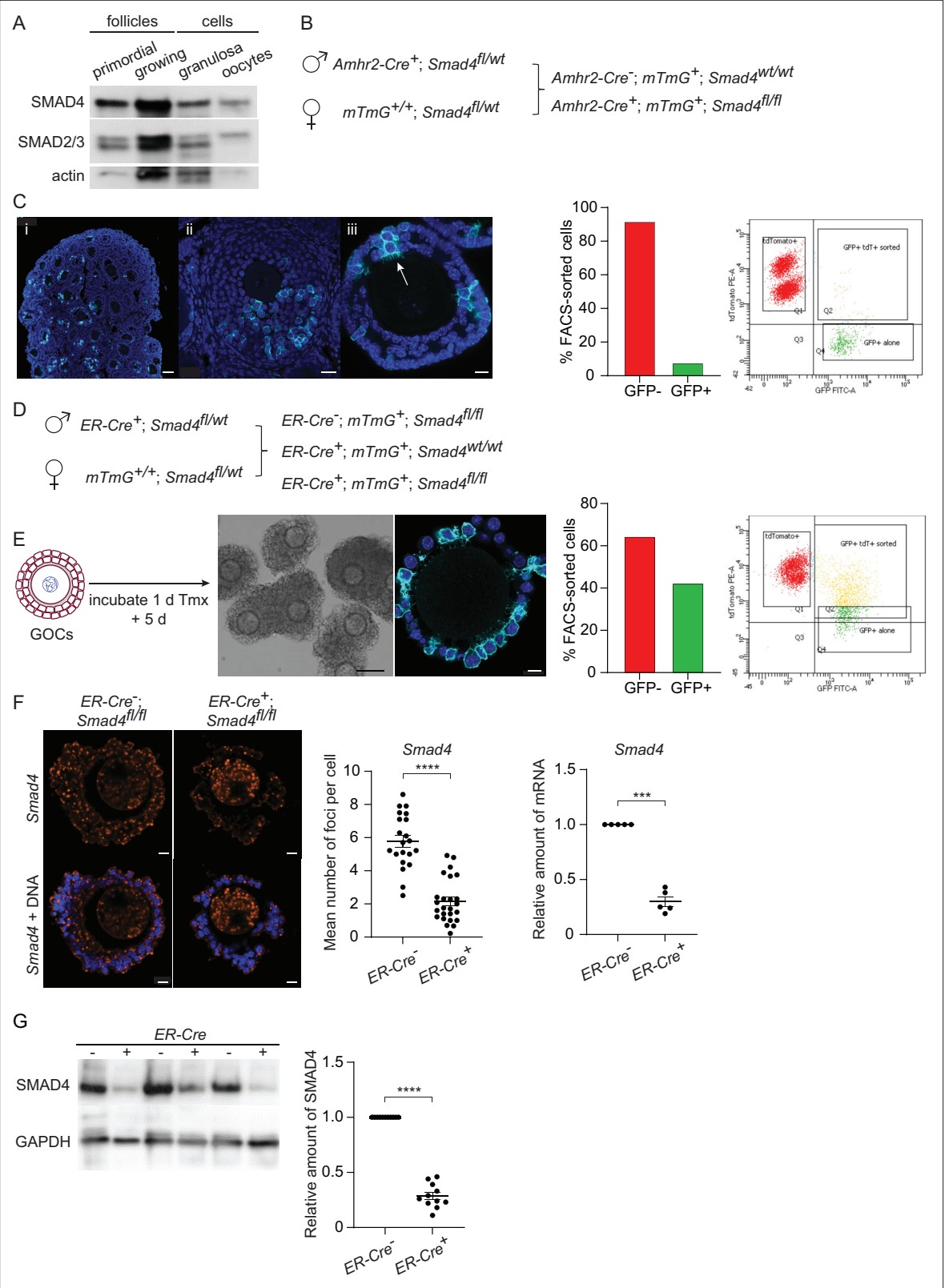

**Figure 1.** SMAD4 is efficiently depleted in granulosa cells of granulosa cell-oocyte complexes (GOCs) in vitro using tamoxifen-induced Cre recombinase. (**A**) Immunoblot of primordial follicles, growing follicles, and granulosa cells and oocytes isolated from antral follicles, probed using anti-SMAD4 and anti-SMAD2/3. (**B**) Breeding scheme to delete *Smad4* in the granulosa cells using *Amhr2-Cre*. The *mTmG* cassette enables Cre recombinase activity to be verified through expression of EGFP. (**C**) Left: Confocal images of histological sections of paraffin-embedded 12-d ovary (**i,**

*Figure 1 continued on next page*

*Figure 1 continued*

ii) and an isolated follicle (iii) stained using an anti-green fluorescent protein (GFP) antibody and DAPI to counterstain DNA. Relatively few granulosa cells are GFP-positive (arrow in iii). Scale bars: 50 µm, 20 µm, 10 µm. Right: Histogram and chart showing the percentage of FACS-sorted GFP + and GFP- granulosa cells from growing follicles in a representative experiment. Granulosa cells expressing both td tomato and GFP and GFP alone were scored as GFP+. (**D**) Breeding scheme to delete *Smad4* in vitro using generate tamoxifen-inducible *ER-Cre*. The *mTmG* cassette enables Cre recombinase activity to be verified through expression of EGFP. (**E**) Left: Schematic illustrating timeline of GOC exposure to tamoxifen and subsequent incubation. Center: Bright-field image of GOCs following incubation and confocal image of a representative GOC stained using anti-GFP antibody and DAPI to counterstain DNA. Scale bar = 100 µm and 10 µm, respectively. Right: Bar chart showing the percentage of FACS-sorted GFP + and GFP- granulosa cells from GOCs in a representative experiment. Granulosa cells expressing td tomato and GFP and GFP alone are counted as GFP+. (**F**) Left: Confocal image of representative GOCs stained with single-molecule RNA FISH probe for *Smad4* exon 8 and DAPI to counterstain DNA. Right: Mean and SEM of the average number of *Smad4* mRNA foci per cell (left) and relative mRNA amounts of *Smad4* quantified using qRT-PCR (right) in granulosa cells five days after tamoxifen treatment. Number of foci: n=26 (*ER-Cre*⁻); n=21 (*ER-Cre*⁺). Relative mRNA: n=5 for both genotypes. (**G**) Immunoblot and plot of mean and SEM of SMAD4 protein content in granulosa cells from GOCs five days after tamoxifen treatment. A. n=11 for both genotypes. Statistical analysis using t-test (F, left) and one-sample t-test (F, right, and G), ***p<0.001; ****p<0.0001.

Because TZPs arise from both the innermost and more distal granulosa cell layers (*Baena and Terasaki, 2019*), it was not possible to reliably determine how many granulosa cells contributed to the GFP-positive TZPs in a GOC. The total number of phalloidin-stained TZPs – irrespective of whether they were GFP-positive – that projected to the oocyte in equatorial optical sections of GOCs from control (*Smad4^{wt/wt}*) and experimental (*Smad4^{fl/fl}*) mice were, therefore, counted (*Figure 2A and B*). TZP number was normalized to the diameter of the corresponding oocyte, although the mean oocyte diameter did not differ between the groups (*Figure 2B*), We observed a 20.4+/-0.05% reduction in the number of TZPs in the GOCs from the *Smad4^{fl/fl}* mice (n=63 [control], 59 [experimental], p=0.002).

A limitation of quantifying TZPs in intact GOCs is that the granulosa cells, depending on their number and spatial distribution, can variably impair their resolution and discrimination even using confocal microscopy. Therefore, the number of TZPs was also counted after mechanically removing the granulosa cells from the GOCs. This analysis revealed a striking 51.1+/-0.04% reduction in the normalized number of TZPs in the GOCs from the *Smad4^{fl/fl}* mice (n=44 [control], 47 [experimental], p<0.0001) (*Figure 2C and D*). As observed using intact GOCs, the mean oocyte diameter did not differ (*Figure 2D*). To verify the difference in TZP-number, we also compared *ER-Cre*⁺; *Smad4^{fl/fl}* to a different control genotype, *ER-Cre*⁻; *Smad4^{fl/fl}* mice. A quantitatively similar 43.6+/-0.04% decrease in TZPs within the *ER-Cre*⁺ population ( +=164 [control], 172 [experimental], p<0.0001) (*Figure 2D*) was observed following tamoxifen treatment and culture. Together, these results indicate that depletion of SMAD4 in the granulosa cells is associated with a decrease in the number of TZPs.

TZPs provide the physical platform that enables essential gap junctional communication between the granulosa cells and the oocyte. To test whether the reduction in TZP number in SMAD4-depleted granulosa cells impaired this communication, we used a fluorescence recovery after photobleaching (FRAP) assay. Control and experimental GOCs were treated with tamoxifen and incubated as above. After five days, they were exposed to calcein-AM, a fluorescent dye that can pass through gap junctions. The fluorescence in the oocyte of each GOC was bleached using the laser of a confocal microscope and its recovery was then recorded. As shown in *Figure 2E*, which plots the fluorescence in the oocyte as a function of time after photobleaching and at the end of the recording period normalized to the initial fluorescence, we observed no difference between groups in the rate of transfer of fluorescence from the granulosa cells to the oocyte. Thus, despite the decreased number of TZPs, depletion of SMAD4 did not detectably impair gap junctional communication between the granulosa cells and the oocyte.

As TZPs project not only from the cells immediately adjacent to the zona pellucida but also from those in more distal layers, it was possible that the reduced number of TZPs in *ER-Cre*⁺; *Smad4^{fl/fl}* complexes reflected an impaired ability of the granulosa cells to proliferate. *ER-Cre*⁺; *Smad4^{wt/wt}* and *ER-Cre*⁺; *Smad4^{fl/fl}* GOCs were incubated for 5 hr in the presence of the thymidine analogue, EdU, added five days after the tamoxifen exposure. No difference between the groups in the fraction of EdU-positive cells was observed (*Figure 2F*), however, implying that the reduced number of TZPs was not due to impaired proliferation of the granulosa cells.

We then focused our analysis on individual granulosa cells. GOCs where individual GFP-positive cells could be readily identified and the TZPs were easily attributable to a particular cell (*Figure 3A*) were selected, serial optical sections were obtained using confocal microscopy, and three-dimensional

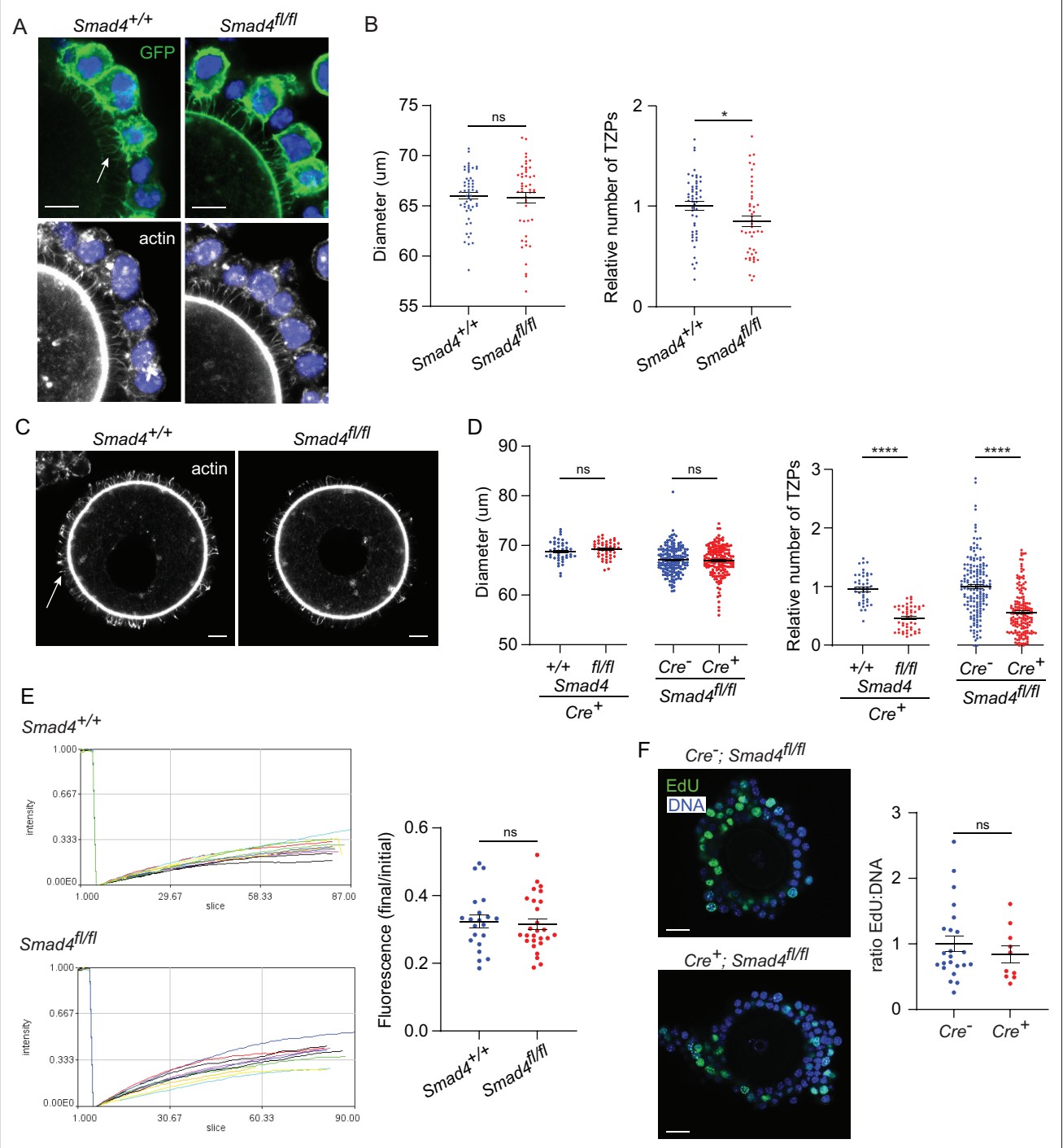

**Figure 2.** Depletion of SMAD4 reduces transzonal projection (TZP)-number but does not affect granulosa cell communication with the oocyte or proliferation. (**A**) Confocal images of granulosa cell-oocyte complexes (GOCs) five days after tamoxifen treatment stained with anti-green fluorescent protein (GFP), phalloidin to visualize F-actin and DAPI to counterstain DNA. Arrow shows GFP-labelled TZPs. Scale bar = 20 μm. (**B**) Mean and SEM of oocyte diameter and TZP number five days after tamoxifen treatment. TZP number in each GOC is normalized to the oocyte diameter. n=63 (+/+), 59 (fl/fl). (**C**) Confocal image of a denuded GOC five days after tamoxifen treatment and removal of the granulosa cells, stained with anti-GFP, phalloidin to visualize F-actin and DAPI to counterstain DNA. Arrow shows phalloidin-labelled TZPs that remain embedded in the zona pellucida. Scale bar = 20 μm. (**D**) Mean and SEM of oocyte diameter and TZP number five days after tamoxifen treatment in denuded GOCs. TZP number for each specimen is normalized to the oocyte diameter. For the groups where all mice were *Cre*⁺: n=44 (+/+), 47 (fl/fl). For the groups where all mice were fl/fl: n=164 (*Cre*⁻), 172 (*Cre*⁺). (**E**) Left: Time course of fluorescence recovery in oocyte of GOCs following bleaching. Each line represents an individual GOC. Right: Mean and SEM of the ratio of the final to pre-bleach fluorescence in each GOC. n=21 (+/+); 27 (fl/fl). (**F**) Left: Confocal image of a GOC after tamoxifen

*Figure 2 continued on next page*

*Figure 2 continued*

treatment followed by treatment with EdU and visualization. DNA was stained using DAPI. Scale bar = 20 µm. Right: Mean and SEM of the ratio of Edu-labelled cells to the total number of cells. n=23 (*Cre*⁻); 10 (*Cre*⁺). Statistical analysis using t-test, *p<0.05; ****p<0.0001; ns, not significant.

images were reconstructed using Imaris software. *Figure 3B* shows one such reconstruction, where GFP-positive granulosa cells are pseudo-coloured in green. We observed a 25% decrease in the mean volume of *Smad4^{fl/fl}* granulosa cells compared to wild-type cells (323.2+/-11.2 vs 242.6+/-10.0; n=293, 322; p<0.0001), but no difference between genotypes in the sphericity of the cells (*Figure 3C*). The TZPs in a subset of the GFP-positive granulosa cells were then analyzed (*Figure 3D*). In this subset, there was a 15% difference in mean cell volume (392.2+/-20.8 vs 334.0+/-21.2; n=55, 63) which did not reach statistical significance (p=0.054). However, whereas wild-type cells harbored a mean of 15.8+/-0.87 (n=55) TZPs per cell, this number was reduced by 35% in *Smad4^{fl/fl}* cells to 10.5+/-0.71 (n=65, p<0.0001). As compared to those of wild-type cells, the TZPs of the *Smad4^{fl/fl}* cells were about

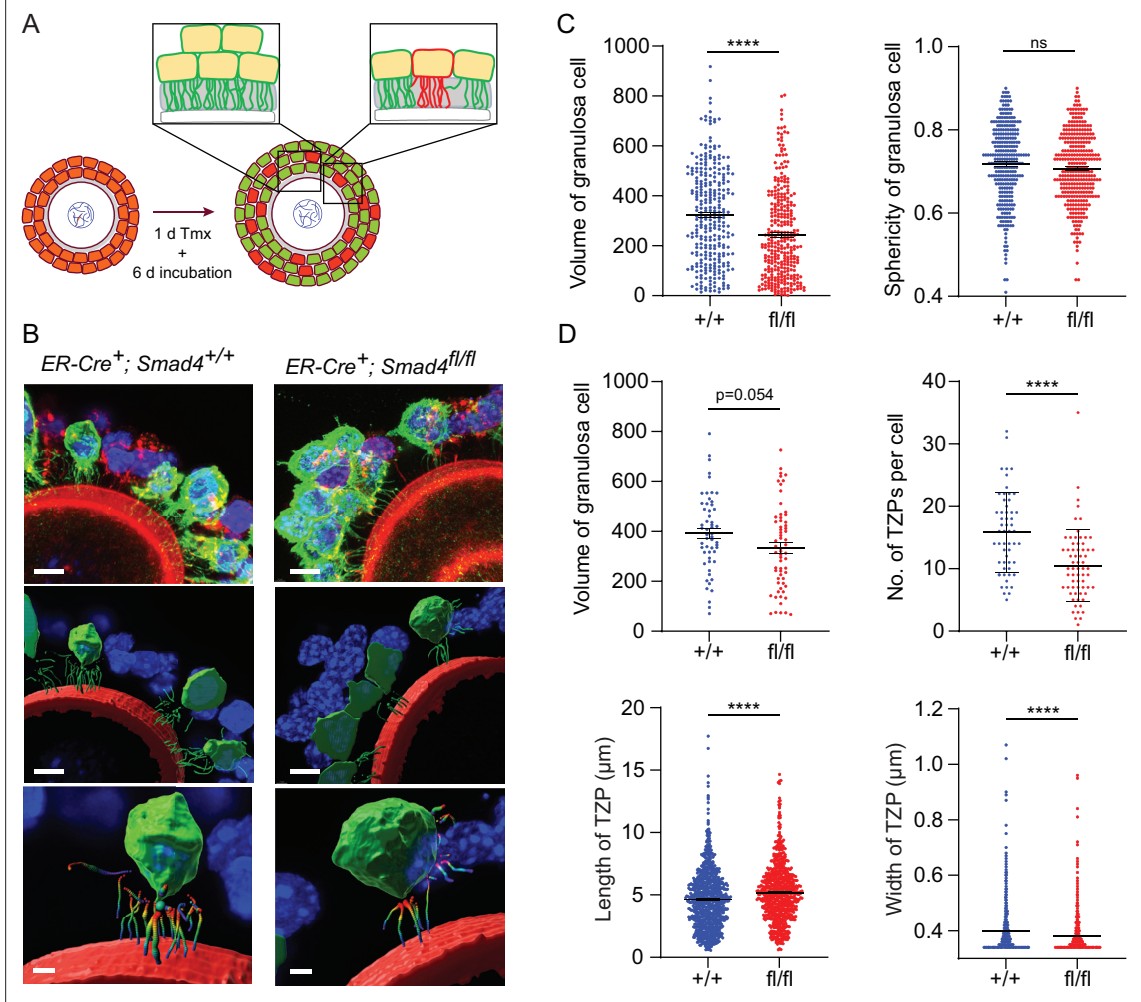

**Figure 3.** Depletion of SMAD4 in granulosa cell-oocyte complexes (GOCs) reduces the number of transzonal projections (TZPs) per granulosa cell and alters their morphometric parameters. (**A**) Schematic illustrating the experimental protocol. green fluorescent protein (GFP+) granulosa cells in clusters were used for calculations in C and individual GFP + granulosa cells were used for calculations in D. (**B**) Three-dimensional reconstruction of a GOC, showing granulosa cell bodies, TZPs, and oocyte cortex. Serial optical confocal sections were processed using Imaris software. GOCs were stained using anti-GFP (green), phalloidin (red), and DAPI (blue). Scale bar = 10 µm (upper, middle), 3 µm (lower). (**C**) Mean and SEM of volumes and sphericities of individual granulosa cells from a cluster and those selected for TZP analyses. Volume: n = 293 (+/+), 322 (fl/fl); sphericity: n = 293 (+/+), 322 (fl/fl). (**D**) Mean and SEM of volumes of individual granulosa cells selected for TZP analysis, number of TZPs per cell, TZP length, and width. Volume: n = 55 (+/+), 63 (fl/fl); no. of TZPs: n = 55 (+/+), 65 (fl/fl) length: n = 893 (+/+), 720 (fl/fl); width: n = 870 (+/+), 719 (fl/fl). Statistical analysis using t-test, ****p<0.0001; ns, non-significant.

10% longer (4.64+/-0.08 μm vs 5.19+/-0.09 μm; n=893, 720; p<0.0001) and very slightly narrower (0.40+/-0.003 μm vs 0.38+/-0.003 μm; n=870, 719; p<0.0001)(*Figure 3E*). We did not detect other differences between the genotypes in TZP morphology. These results obtained using reconstructed 3-dimensional images of GFP-positive cells establish that the number of TZPs is reduced in individual SMAD4-depleted granulosa cells.

A limitation of analyzing TZPs in GFP-positive cells within GOCs is that this strategy cannot distinguish TZPs that were generated in a low-SMAD4 environment (i.e. after the Cre recombinase was expressed) from those generated earlier in a normal-SMAD4 environment. To ensure that we examined only TZPs that were generated after Cre had been expressed, we used our previously published reaggregation assay (*El-Hayek et al., 2018a*; *Figure 4A*). GOCs obtained from mid-growth stage follicles of *ER-Cre; mTmG* mice carrying the appropriate *Smad4* alleles were exposed to tamoxifen for one day and then incubated for an additional four days. The granulosa cells were then separated from the oocyte and combined with an oocyte that did not carry the *mTmG* gene. These reaggregated GOCs were incubated for an additional six days, then fixed and stained using anti-GFP (*Figure 4B*).

When the entire population of granulosa cells in the reaggregated GOCs was analyzed, there was a ~30% reduction in mean volume in the *Smad4^fl/fl* cells as compared to *Smad4^wt/wt* cells (394.5+/-18.8 vs 279.8+/-16.6; n=185, 159; p<0.0001) and no difference in cell sphericity (*Figure 4C*), as observed using intact GOCs. There was no difference between genotypes, however, in mean volume among the cells whose TZPs were characterized (*Figure 4D*). Within this group, the mean number of TZPs per cell was reduced by ~50%, from 14.2+/-2.0 (n=16) in the *Smad4^wt/wt* cells to 6.84+/-1.1 (n=25) in the *Smad4^fl/fl* cells (*Figure 4D*, p=0.001). A ~20% increase in the mean length of the TZPs of *Smad4^fl/fl* cells (3.71+/-0.17 vs 4.40+/-0.28; n=174, 154; p=0.03) was also detected, as in intact GOCs, but no difference in the mean width (*Figure 4E*).

The increased mean length of the TZPs of *Smad4^fl/fl* cells compared to *Smad4^wt/wt* cells prompted us to analyze the trajectory of individual TZPs. We traced their path from their point of origin to their tip, then measured the distance between the point of origin and the oocyte and subtracted from this the distance between the tip and the oocyte. A large positive difference indicates that the tip of the TZP is relatively close to the oocyte; conversely, a small positive or a negative difference indicates that that tip is relatively far from the oocyte (*Figure 4F*). This analysis revealed that the mean difference was 50% greater in the *Smad4^fl/fl* cells compared to *Smad4^wt/wt* cells (1.61+/-0.16 vs 2.40+/-0.21; n=174, 157; p=0.003) (*Figure 4G*). This indicates that the tips of the TZPs of the *Smad4^fl/fl* cells were, on average, closer to the oocyte than those of the *Smad4^wt/wt* cells. We also found that the proportion of TZPs that were oriented away from the oocyte trended towards being greater in the *Smad4^wt/wt* cells, although the difference did not reach statistical significance (*Figure 4G*). Taken together, the trajectory analyses provide additional evidence that the TZPs of *Smad4^fl/fl* cells differ from those of *Smad4^wt/wt* cells.

The observation that TZP-number was modestly reduced when intact GOCs were analyzed, yet markedly reduced when oocytes from which the granulosa cells had been stripped were analyzed, suggested that the attachment between the TZPs and oocyte might be weakened when SMAD4 was depleted. To evaluate this, the amount of N-cadherin, which is the principal cell adhesion molecule that has been identified in granulosa cells, was compared between *Smad4^wt/wt* and *Smad4^fl/fl* granulosa cells. Consistent with the previous results (see *Figure 1E*), SMAD4 was reduced by 73.5+/-0.05% in the *Smad4^fl/fl* granulosa cells (*Figure 5A*, n=6, p<0.0001). Strikingly, N-cadherin was reduced by 52.7+/-0.04% (n=6, p=0.0004) in the *Smad4^fl/fl* granulosa cells (*Figure 5A*). Moreover, Notch2, a transmembrane receptor expressed in granulosa cells that associates with membrane-bound Jagged1 expressed on the oocyte, was reduced by 52.6+/-0.05% (n=5; p=0.0003) in these cells (*Figure 5B*). In contrast to the protein, however, there was no detectable reduction in the amount of encoding mRNA in either case as assayed using qRT-PCR, although *Smad4* was reduced in the same samples (*Figure 5C*). Thus, a reduction in SMAD4 in granulosa cells is associated with a reduction in certain transmembrane proteins, including those responsible for cell adhesion.

As we previously reported that depletion of GDF9, which is secreted by the oocyte and signals in the granulosa cells through SMAD4, was associated with a reduction in mRNAs encoding structural components of TZPs – *Myo10*, *Fscn1*, and *Daam1* (*El-Hayek et al., 2018a*), we measured the quantity of these mRNAs using qRT-PCR. We found that they were not depleted in the *Smad4^fl/fl* granulosa cells compared to the *Smad4^wt/wt* cells (*Figure 5D*). This implies either that the SMAD4 which remained

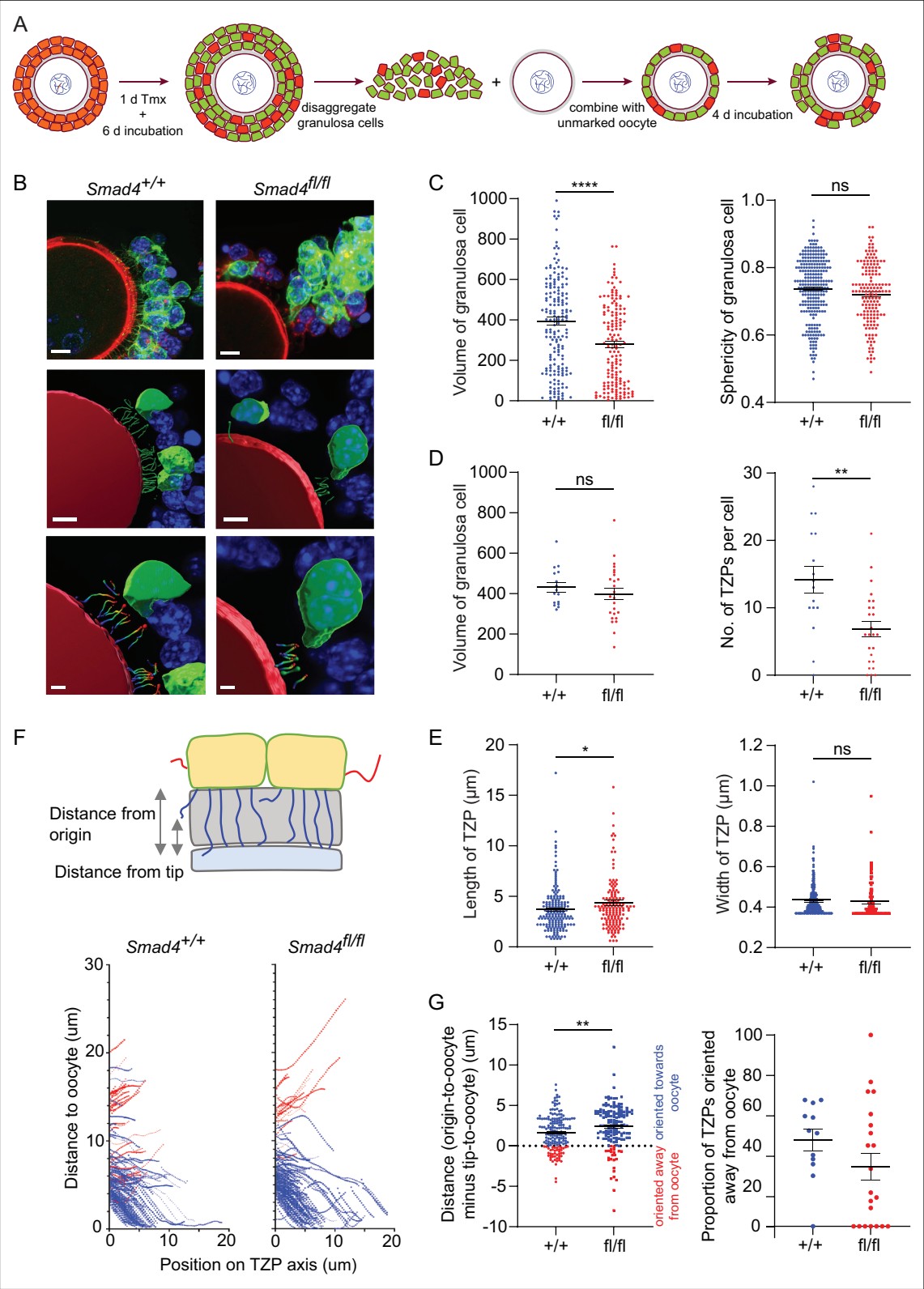

**Figure 4.** Depletion of SMAD4 in reaggregates reduces the number of newly generated transzonal projections (TZPs) and alters their morphometric parameters. (**A**) Schematic illustrating the reaggregation procedure. (**B**) Three-dimensional reconstruction of a reaggregate, showing granulosa cell bodies, TZPs, and oocyte cortex. Serial optical confocal sections were processed using Imaris software. Granulosa cell-oocyte complexes (GOCs) were stained using anti-green fluorescent protein (GFP) (green), phalloidin (red), and DAPI (blue). Scale bar = 10 μm (upper), 3 μm (lower). (**C**) Mean and

*Figure 4 continued on next page*

*Figure 4 continued*

SEM of volumes and sphericities of individual granulosa cells. Volume: n = 185 (*+/+*), 159 (*fl/fl*); sphericity: n = 185 (*+/+*), 159 (*fl/fl*). (**D, E**) Mean and SEM of volumes of individual granulosa cells, number of TZPs per cell, TZP length, and TZP width. Volume: n = 16 (*+/+*), 25 (*fl/fl*); no. of TZPs: n = 16 (*+/+*), 25 (*fl/fl*) length: n = 174 (*+/+*), 154 (*fl/fl*); width: n = 316 (*+/+*), 169 (*fl/fl*). (**F**) Upper: Schematic illustrating how the distance from the origin and the tip, respectively, of TZPs (blue) and non-TZP projections (red) to the oocyte surface was determined. Lower: TZP trajectories of 3D-reconstructed TZPs, representing the distance from origin relative to the position on the TZP axis. (**G**) Left: Mean and SEM of the difference between the distance from the origin and the tip of individual TZPs to the oocyte surface. Blue symbols (>0) represent TZPs oriented towards the oocyte; red symbols (<0) represent non-TZPs oriented away from the oocyte. n = 174 (*+/+*), 157 (*fl/fl*). Right: Mean and SEM of the proportion of TZPs in individual cells that were oriented away from the oocyte. n = 12 (*+/+*), 21 (*fl/fl*). Statistical analysis using t-test, *p<0.05; **p<0.01; ****p<0.0001; ns, non-significant.

The online version of this article includes the following figure supplement(s) for figure 4:

**Figure supplement 1.** Segmentation pipeline of green fluorescent protein (GFP)-positive granulosa cells.

was sufficient to induce expression of these genes or the GDF9 regulates their expression through a SMAD4-independent pathway.

## Discussion

We have examined the role of SMAD4 in establishing the TZPs that connect the granulosa cells to the oocyte. Using a tamoxifen-inducible Cre recombinase strategy, we were able to reduce SMAD4 in the granulosa cells to about 30% of its wild-type level. This reduction was associated with a decrease ranging from 20–50% in the number of TZPs in the granulosa cells assessed as a population (i) in intact GOCs and (ii) after removal of the granulosa cell bodies to increase optical resolution, and as individual cells (iii) in intact GOCs and (iv) in reconstructed complexes. Previous studies where SMAD4, or both SMAD2 and SMAD3, was depleted from granulosa cells described a loose or loss association of

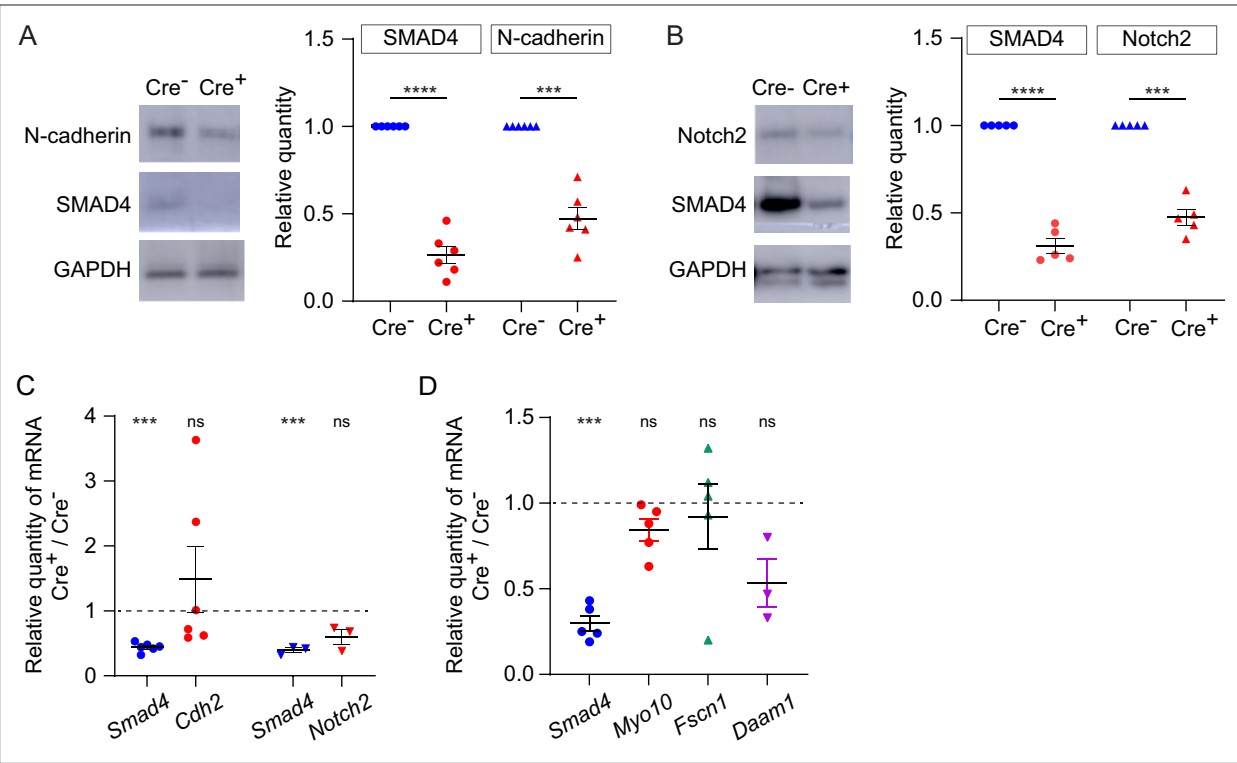

**Figure 5.** Expression of transmembrane N-cadherin and Notch proteins is reduced in SMAD4-depleted granulosa cells. (**A**) Immunoblot and graph showing mean and SEM of SMAD4 and N-cadherin in granulosa cells. Protein quantities are normalized to amounts in ER-Cre⁻ cells. n=6. (**B**) As in A, showing Notch2. n=5. (**C**) Mean and SEM of relative mRNA amounts of *Smad4*, *Cdh2* (encoding N-cadherin), and *Notch2* in granulosa cells purified from granulosa cell-oocyte complexes (GOCs) five days after tamoxifen treatment. mRNA quantities are normalized to amounts in ER-Cre⁻ cells. n=6 (*Cdh2*), 3 (*Notch2*). (**D**) As in C, showing *Smad4*, *Myo10*, *Fscn1,* and *Daam1*. n=5, except for *Daam1* (n=3). Statistical analysis using one-sample t-test, ***p<0.001; ****p<0.0001; ns non-significant.

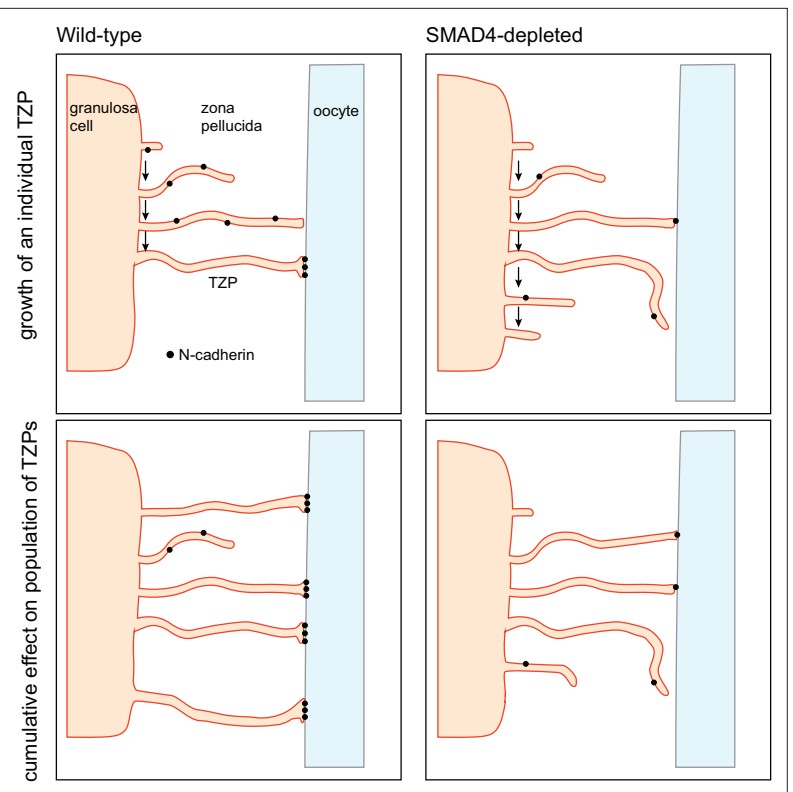

**Figure 6.** Model for regulation of the size of the transzonal projection (TZP) population. Upper: Dynamics of an individual TZP. When growing TZP contacts the surface of the oocyte, it becomes stably attached if there is sufficient N-cadherin, as in wild-type cells. In SMAD4-depleted cells, there is insufficient N-cadherin to stabilize the attachment. The TZP may continue to elongate but eventually becomes retracted. Lower: Dynamics of the population of TZPs. Owing to increased stability of attachment of individual TZPs to the oocyte surface, the total number of TZPs becomes larger in wild-type cells than in SMAD4-depleted cells.

the cumulus cells with the oocyte in some follicles (*Pangas et al., 2006*; *Li et al., 2008*). Moreover, as described in the Introduction, blocking the production of GDF9 by the oocyte leads to a decrease in the number of TZPs (*Carabatsos et al., 1998*; *El-Hayek et al., 2018a*). Together these observations suggest that SMAD4 regulates the size of the population of TZPs in growing follicles.

The population of TZPs presumably reflects the net result of the generation of new TZPs and the retraction of existing TZPs, as is the case for filopodia (*Gallop, 2020*; *Heckman and Plummer, 2013*; *Mattila and Lappalainen, 2008*). SMAD4 could thus promote either or both processes. The observation that GFP-expressing granulosa cells in reaggregated complexes generated new TZPs implies that SMAD4 is not required for this process, although it does not rule out a role in increasing the rate of generation. A clue to the function of SMAD4 may lie in the observation that the decrease in the number of TZPs in SMAD4-depleted cells was quantitatively greater when the granulosa cell bodies were removed prior to analysis. This suggests that the process of removing the cell bodies also removed the TZPs more easily in the SMAD4-depleted cells than in wild-type cells.

In this context, it is intriguing that N-cadherin, which is thought to stabilize the attachment of the TZP tips to the oocyte surface, was depleted in the mutant cells. We propose that, by increasing the quantity of N-cadherin in the granulosa cells, SMAD4 increases the strength of adhesion of the TZPs to the oocyte surface (*Figure 6*). This in turn would decrease the probability of retraction, thereby increasing the steady-state number of TZPs. This model is consistent with the observation that the non-attached (free-ended) TZPs of SMAD4-depleted cells were on average longer and closer to the oocyte than those of wild-type cells. Specifically, if the ability of the TZPs to establish a stable contact with the oocyte wer impaired in SMAD4-depleted cells, a relatively large proportion might continue to 'search' for an attachment and to lengthen during the searching process.

Mechanistically, some reports have linked SMAD4 to transcription of the *Cdh2*, which encodes N-cadherin (*Kang et al., 2014*; *Yang et al., 2015*), but we did not detect a difference in mRNA quantity between normal and SMAD4-depleted granulosa cells. In view of the reduced quantity in the latter of Notch2, also a membrane-associated protein, it may be speculated that SMAD4 might act more broadly to regulate the expression of genes whose products maintain the normal configuration of the granulosa cell plasma membrane. An alternative explanation is raised by observations that granulosa-cell depletion of SMAD4 is associated with the presence of follicles containing luteinized granulosa cells enclosing an entrapped oocyte and increased expression of markers of luteinization, including *Cyp11a1*, *Sfrp4*, *Star,* and *Ptgfr* (*Pangas et al., 2006*). This suggests that the SMAD4-depleted granulosa cells in the GOCs studied here might have become luteinized, and that luteinization in turn caused a reduction in the number of TZPs. We did not detect increased expression of the luteinization markers (unpublished data), suggesting that luteinization did not occur under the conditions of our experiments. This difference may arise because we examined GOCs that had been maintained in vitro, whereas the previous study examined follicles in intact ovaries of prepuberal mice following the injection of an FSH analogue.

Regardless of the mechanism linking SMAD4 to TZP number, it is noteworthy that the number of TZPs that project from an individual granulosa cells – about 15 in this study and about 40 in a previous study where more advanced (antral) follicles were examined (*Baena and Terasaki, 2019*) – considerably exceeds the number of filopodia that are present on individual blastomeres of mouse embryos (5-6) (*Fierro-González et al., 2013*) or on chick embryonic cells (~4) (*Pröls et al., 2015*). SMAD4 may, therefore, act in granulosa cells to amplify an innate ability to generate filopodia, which become TZPs and increase the efficiency of communication between the growing germ cell and its somatic microenvironment.

## Methods

**Key resources table**

| Reagent type (species) or resource | Designation | Source or reference | Identifiers | Additional information |
|---|---|---|---|---|
| gene (*M. musculus*) | Smad4 | GenBank | Gene ID: 17128 | |
| strain, strain background (*M. musculus*) | Smad4$^{tm2.1Cxd}$/J | Jackson Laboratories | strain 017462 | Supplied by Dr. Daniel. J. Bernard, McGill University |
| strain, strain background (*M. musculus*) | Amhr2$^{tm3(cre)Bhr}$ | MMRRC | strain 014245-UNC | Supplied by Dr. Makoto Nagano, McGill University |
| strain, strain background (*M. musculus*) | B6.129-Gt(ROSA)26Sor$^{tm1(cre/ERT2)Tyj}$/J | Jackson Laboratories | strain 008463 | |
| strain, strain background (*M. musculus*) | B6.129(Cg)-Gt(ROSA)26Sor$^{tm4(ACTB-tdTomato,-EGFP)Luo}$/J | Jackson Laboratories | strain 007676 | |
| antibody | anti-SMAD2/3 (Rabbit monoclonal) | Cell Signaling Technology | 8685 RRID: AB_10889933 | 1:1000 (immunoblot) |
| antibody | anti-SMAD4 (Rabbit polyclonal) | Sigma | HPA019154 RRID: AB_1853480 | 1:1000 (immunoblot) |
| antibody | anti-N-cadherin (Mouse monoclonal) | BD Biosciences | B610920 RRID: AB_2077527 | 1:1000 (immunoblot) |
| antibody | anti-Notch2 (Rabbit polyclonal) | Abcam | Ab8926 RRID: AB_2267338 | 1:1000 (immunoblot) |
| antibody | anti-GFP (Rabbit polyclonal) | Invitrogen | A11122 RRID: AB_2307355 | 1:100 (immunofluorescence, immunohistochemistry) |

*Continued on next page*

*Continued*

| Reagent type (species) or resource | Designation | Source or reference | Identifiers | Additional information |
|---|---|---|---|---|
| antibody | anti-rabbit IgG-HRP (Goat polyclonal) | Promega | W4011 RRID: AB_430833 | 1:5000 (immunoblot) |
| antibody | anti-mouse IgG-HRP (Goat polyclonal) | Promega | W4021 RRID: AB_430834 | 1:5000 (immunoblot) |
| antibody | anti-rabbit IgG-Alexa 488 (Donkey polyclonal) | Invitrogen | A21206 RRID: AB_2535792 | 1:100 (immunofluorescence, immunohistochemistry) |
| sequence-based reagent | Primer for polymerase chain reaction (PCR) genotyping | Sigma | Smad4 – wild-type | 5':GGGCAGCGTAGCATATAAGA 3':GACCCAAACGTCACCTTCAG Predicted size: 390 nt |
| sequence-based reagent | Primer for PCR genotyping | Sigma | Smad4 – floxed | 5':GGGCAGCGTAGCATATAAGA 3':GACCCAAACGTCACCTTCAG Predicted size: 450 nt |
| sequence-based reagent | Primer for PCR genotyping | Sigma | Smad4 – recombined | 5':TAAGAGCCACAGGGTCAAGC 3':GACCCAAACGTCACCTTCAG Predicted size: 450 nt |
| sequence-based reagent | Primer for PCR genotyping | Sigma | Amhr2-Cre | 5':CTCTGGTGTAGCTGATGATC 3':TAATCGCCATCTTCCAGCAG Predicted size: 380 nt |
| sequence-based reagent | Primer for PCR genotyping | Sigma | Rosa26$^{CreERT2}$ | 5':CGTGATCTGCAACTCAGTC 3': AGGCAAATTTTGGTGTACGG Predicted size: 150 nt |
| sequence-based reagent | Primer for reverse-transcription (RT)-PCR | Sigma | Smad4 | 5':TCACAATGAGCTTGCATTCC 3':CCATCCACAGTCACAACAGG Predicted size: 143 nt |
| sequence-based reagent | Primer for reverse-transcription (RT)-PCR | Sigma | Cdh2 (N-cadherin) | 5':CCAGGAAAAGTGGCAGGTAG 3':CACTGTGCTTGGCAAGTTGT Predicted size: 121 nt |
| sequence-based reagent | Primer for reverse-transcription (RT)-PCR | Sigma | Notch2 | 5':GACCCTATCCTACCCTCTAGTG 3':AGCAGGATGAAGAACAGGATG Predicted size: 103 nt |
| sequence-based reagent | Primer for reverse-transcription (RT)-PCR | Sigma | Myo10 | 5':TCCAGACAGACTATGGGCAG G 3':GGAAGCCATGTCGTCCACG Predicted size: 109 nt |
| sequence-based reagent | Primer for reverse-transcription (RT)-PCR | Sigma | Fscn1 | 5':AGAACGCCAGCTGCTACTTT 3':CGAGGAATCACTACCCACCG Predicted size: 331 nt |
| sequence-based reagent | Primer for reverse-transcription (RT)-PCR | Sigma | Daam1 | 5':GCGGCTGCTCAGAGTATAGAAA 3':AAACATGGCTTCCCTGTGTTTG Predicted size: 273 nt |
| sequence-based reagent | Primer for reverse-transcription (RT)-PCR | Sigma | Rpl19 | 5':GAAATCGCCAATGCCAACTC 3':CTTCCCTATGCCCATATGCC Predicted size:147 nt |
| sequence-based reagent | RNA in situ hybridization probe | Advanced Cell Diagnostics | 857571 | Smad4, exon 8 |
| sequence-based reagent | RNA in situ hybridization probe | Advanced Cell Diagnostics | 310043 | *Bacillus subtilis* DapB |
| commercial assay or kit | Click-iT EdU Alexa Fluor 488 Imaging Kit | Molecular Probes | C10337 | In situ detection of DNA replication |
| commercial assay or kit | BaseScope Assay | Advanced Cell Diagmostics | 322337 | In situ detection of RNA |
| chemical compound, drug | 4-hydroxytamoxifen | Sigma | T176 | 1 µg/ml |

*Continued*

| Reagent type (species) or resource | Designation | Source or reference | Identifiers | Additional information |
|---|---|---|---|---|
| chemical compound, drug | Phalloidin (Alexa Fluor 568) | Thermo Fisher | A12380 | 1:100 |
| chemical compound, drug | EdU | Thermo Fisher | C10337 | 100 μM |
| software, algorithm | Imaris | Bitplane | 9.7.2 | Image analysis |

## Mouse strains

All experiments were performed in compliance with the regulations and policies of the Canadian Council on Animal Care and were approved by the Animal Care Committee of the Research Institute of the McGill University Health Centre (RI-MUHC, protocol 7783). CD-1 mice were obtained from Charles River (St-Constant, QC). The following transgenic strains were obtained from commercial suppliers: B6.129-*Gt(ROSA)26Sor*^*tm1(cre/ERT2)Tyj*/J (referred to as *ER-Cre* in the text) carrying a tamoxifen-inducible sequence encoding Cre recombinase under the control of the *Rosa26* regulatory elements (Jackson Laboratories, Bar Harbor, ME, strain 008463); B6.129(Cg)-*Gt(ROSA)26Sor*^*tm4(ACTB-tdTomato,-EGFP)Luo*/J (Jackson, strain 007676; hereafter *mTmG*) carrying sequences encoding a membrane-targeted tdTomato and membrane-targeted EGFP; Cre-mediated recombination of *loxP* sites switches expression from RFP to EGFP. *Smad4*^*tm2.1Cxd*/J (hereafter *Smad4*^*fl*) carrying *loxP* sites flanking exon 8 (Jackson, strain 017462) was kindly supplied by Dr. Daniel Bernard (McGill University). *Amhr2*^*tm3(cre)Bhr* (hereafter *Amhr2-Cre*) carrying a sequence encoding Cre recombinase under the control of the promoter of the Anti-Mullerian hormone receptor 2 gene was kindly supplied by Dr. Makoto Nagano (McGill University).

To generate *Amhr2-Cre*^+; *Smad4*^*fl/fl* females carrying an *mTmG* allele, *Smad4*^*fl/wt*; *mTmG*^*+/+* females were crossed with *Amhr2-Cre*^+; *Smad4*^*fl/wt* males. To generate *ER-Cre*^+; *Smad4*^*fl/fl* females carrying an *mTmG* allele, *Smad4*^*fl/wt*; *mTmG*^*+/+* females were crossed with *ER-Cre*^+; *Smad4*^*fl/wt* males. *ER-Cre*^-; *mTmG*^+; *Smad4*^*fl/fl* females and *ER-Cre*^+; *mTmG*^+; *Smad4*^*wt/wt* females served as controls. Genotyping of the offspring was done using the EZ Tissue/Tail DNA Isolation Plus PCR kit (EZ Bioresearch). After DNA extraction from tail snips, PCR was used to detect wild-type, floxed, and recombined *Smad4* and the presence of *ER-Cre* and *Amhr2-Cre* alleles.

## Tissue embedding and immunohistochemistry

Sections of paraffin-embedded ovaries were deparaffinized using xylene, rehydrated in a decreasing ethanol series (100, 90, 70%) for 3 min each, and then placed in Milli-Q water for 5 min. Slides were heated intermittently in a microwave during 20 min in 0.1 M citrate buffer solution at pH 6.0, washed in PBS for 10 min, and blocked with CAS universal blocking solution (Thermo Fisher, 008120) for 30 min at room temperature to reduce nonspecific binding. After overnight incubation with primary antibody, slides were washed three times in PBS at room temperature and then incubated for 1 hr at room temperature with secondary antibody with DAPI. Slides were washed twice in PBS for 5 min and mounted on a drop of VECTASHIELD solution (Vector Laboratories, H-1200–10). Images were acquired using a Zeiss LSM 880 confocal microscope.

## Collection, culture, and tamoxifen treatment of GOCs

GOCs were obtained as previously described (*El-Hayek et al., 2018b*; *O'Brien et al., 2003*). Briefly, ovaries of 10- to 12-day-old mice were dissected from the ovarian bursa. Follicles were mechanically isolated by carefully tearing the ovaries apart and using fine needles to separate the follicles from each other in HEPES-buffered minimal essential medium (MEM, pH 7.2, Life Technologies, Burlington, ON) (MEM-H). A Pipetman P1000 was used to carefully pick up and transfer the individual follicles to dishes containing fresh medium. GOCs were obtained by incubating follicles in the presence of collagenase (10 μg/ml; Cedarlane, Burlington, ON) and DNase I (10 μg/ml; Sigma) for 5 min at 37 °C in air. GOCs were collected and briefly washed in $NaHCO_3$-buffered MEM (MEM-C) and cultured in four-well plates containing 500 μl of MEM-C supplemented with ITS (10 μg/mL insulin, 5.5 μg/mL transferrin and 5 ng/

mL selenium; Sigma), FSH (10 mIU/ml; EMD Serono, Mississauga, ON), and estradiol (10 nM, Sigma, E2758) in an atmosphere of 5% $CO_2$ in air. To induce Cre-mediated recombination of the *Smad4* allele, GOCs were incubated with 4-hydroxytamoxifen (1 µg/ml, Sigma, T176) in the above medium for 24 hr. They were then transferred to a medium without tamoxifen.

## Immunoblotting

Cell samples were lysed in 10 µl of Laemmli buffer. After denaturation at 95°C for 10 min, proteins were separated using 4–15% precast polyacrylamide gels (BioRad, 4568085) and transferred onto a polyvinylidene fluoride membrane (Amersham, Oakville, ON, Canada) under constant voltage. The membrane was subsequently blocked using 5% BSA, in 0.1% Tween-PBS. The membrane was washed three times in Tween-PBS and incubated with primary antibody overnight at 4°C. After washing for 45 min, the membrane was incubated in secondary antibody conjugated to horseradish peroxidase (Promega) at a dilution of 1:5000 for 1 hr at room temperature. After washing for 30 min, the bound antibody was revealed using ECL+ (Amersham).

## RNA purification and quantitative real-time PCR

RNA was extracted using a Picopure RNA isolation kit (Life Technologies) following the manufacturer's instructions and eluted in 10 µL of the provided elution buffer, as described (*El-Hayek et al., 2018a*). Briefly, SuperScript II Reverse Transcription kit (Life Technologies) was used to generate cDNA. PCR amplification was performed using a Corbett Rotorgene 6000 (Montreal Biotech, Montreal, QC). Each reaction contained 4 µL of EvaGreen Mix (Montreal Biotech), 13 µL of UltraPure DNase/RNase-free distilled water (Life Technologies), 1 µL of 10 mM primers, and 2 µL of cDNA (diluted 1:20 from original stock). Primers were designed using Primer-BLAST (National Institutes of Health) and obtained from Sigma. For each primer pair, a standard curve was generated using serial dilutions of cDNA prepared from ovarian RNA and used to determine the efficiency of amplification. Melt-curve analysis and electrophoresis of amplified products confirmed that only a single product of the expected size was generated. Data was analyzed using software provided by the manufacturer. Relative quantities of amplified product were calculated according to 2-ΔΔCT method, using *Rpl19* for normalization.

## Immunofluorescence

GOCs and denuded oocytes were fixed for 15 min in freshly prepared 2% (w/v) para-formaldehyde (Fisher Scientific 04042) in PBS (pH 7.2) containing 0.1% Triton X-100 (ACROS 9002-93-1) (PBST), then washed in PBST. The specimens were incubated overnight in primary antibody in PBST at 4°C with gentle agitation, washed twice in PBST, then incubated for 1 hr at room temperature in the secondary antibody as well as phalloidin and DAPI in PBST, then washed in PBST. GOCs or denuded oocytes were placed in a 2 µl drop of PBS containing 0.3% polyvinylpyrrolidone (Sigma, PVP360) on a glass bottom dish covered with mineral oil. Images were acquired using a Zeiss LSM880 confocal microscope. Images were obtained as 16-bit RGB stack files. GOCs were imaged with a Plan Apo 63 x/1.4 oil immersion objective and the following lasers and filters (in parentheses): 488 nm (491–553 nm), 561 nm (571–642) and 405 nm (410–484). Laser strength ranged between 1.5 and 2% and digital gain was adjusted until the first saturated pixels were observed for each channel.

## EdU incorporation and detection

EdU (5-ethynyl-2'-deoxyuridine) detection was performed on GOCs according to the Click-iT EdU Alexa Fluor 488 Imaging Kit manufacturer's instructions (Molecular Probes, C10337). Briefly, cells were incubated in MEM containing 100 µM EdU for 5 hr to allow for incorporation into newly synthesized DNA, then fixed, permeabilized, and incubated in Click-iT cocktail reaction for 30 min. EdU labeling was visualized by confocal microscopy and quantified with ImageJ (NIH).

## In situ hybridization

GOCs were fixed in freshly prepared 2% paraformaldehyde in phosphate-buffered saline (PBS, pH 7) for 15 min at room temperature, dehydrated by passing through a gradient of ice-cold methanol (50%, 75%, 100%) for 3 min each, then sealed and stored at 4°C. The next day, the GOCs were rehydrated using the reverse methanol gradient (75%, 50%, PBS) for 3 min each. Up to 10 GOCs were placed in the middle of a glass microscope slide, encircled by a hydrophobic barrier created

using an Immedge pen, and incubated with protease solution (ACD, 322337) for 20 min. Slides were rinsed three times with PBS and processed as indicated by the manufacturer for the BaseScope Assay (Advanced Cell Diagnostics (ACD), 323971). Briefly, slides were incubated with the *Smad4* probe specific for the exon8 allele (ACD, 857571) or negative control probe against *Bacillus subtilis DapB* (ACD, 310043) at 40°C for 2 hr in a HybEZ oven. Subsequent amplification and detection steps were performed as recommended by the manufacturer and slides were taken for image acquisition using a Zeiss LSM 880 confocal microscope.

## Fluorescence-activated cell sorting

GOCs were dissected from the ovaries of 12–15 d-old *Amhr2-Cre⁺; mTmG⁺; Smad4ᶠˡ/ᶠˡ* mice or were collected after tamoxifen treatment and culture in the case of *ER-Cre+; mTmG +Smad4ᶠˡ/ᶠˡ* mice. GOCs dissected from *Amhr2-Cre⁻* and *ER-Cre⁻* littermates were used as negative controls of EGFP expression. They were transferred to a four-well dish containing Ca-free MEM (Thermo Fisher, 21068028) supplemented with 0.25% trypsin (Thermo Fisher, 25200056), 1 mM EDTA (Sigma), and DNAse I for several minutes. After pipetting up and down for a few times to dissociate the cells, fetal bovine serum was added (4% final concentration) to inactivate the trypsin. Cells were centrifuged at 1500xg for 5 min and supernatant was carefully removed under the dissecting microscope to better visualise the pellet. FACs buffer containing PBS, 1% FBS, 1 mM EDTA, and 1 µg/ml of DAPI was used to resuspend the pellet. Cells were resuspended and passed through a Flowmi 40 µm strainer (Merck, BAH136800040) to remove oocytes and cell clumps, and then kept on ice for sorting. Resulting granulosa cells were analyzed on a BD FACSAria Fusion 8.0.2 sorter at the Immunophenotyping Facility of the RI-MUHC. All events were collected for each sample. A 488 nm excitation laser was used for both Tomato and EGFP excitation since the wavelength of 488 nm falls within the peak excitation of EGFP but only at 25% of peak excitation for tdTomato. Data were analyzed using FlowJo FACS analysis software. Cells were gated for singlets (using forward scatter vs. side scatter area), viability (DAPI scatter vs. side scatter area), and fluorescence (FITC and PE) without compensation.

## Measurement of granulosa cell-oocyte communication using fluorescence recovery after photobleaching (FRAP)

A protocol for FRAP was adapted from our previous study (*El-Hayek and Clarke, 2015*). GOCs were incubated for 15 min in MEM supplemented with calcein-AM and then for 60 min in calcein-free MEM to allow the transfer of calcein from the granulosa cells into oocyte. A circular region of interest of fixed diameter (10 µm, to ensure the bleaching did not extend to the surrounding granulosa cells) was set in the center of the enclosed oocyte and subjected to 60 repetitive bleaches of 50 iterations (1 min total duration) using the argon laser at 50% transmission strength. The intensity of fluorescence in the oocyte and the layer of granulosa cells immediately surrounding it was recorded before bleaching and after the 10ᵗʰ, 20ᵗʰ, 30ᵗʰ, 40ᵗʰ, 50ᵗʰ, and 60ᵗʰ bleaching. Fluorescence was calculated as intensity of fluorescence in granulosa cell layer at each time point divided by the pre-bleach intensity of the layer. As oocytes of all experimental groups were subjected to the same bleaching procedure within the fixed ROI, the loss in fluorescence was further divided by initial pre-bleach intensity of the oocyte normalized to intensity of fluorescence in oocyte at the given time point, in order to correct for differences in bleaching effect in oocytes of different sizes.

## Generation of granulosa cell-oocyte reaggregates

GOCs were obtained from 12 day *ER/Cre⁺; mTmG⁺; Smad4ʷᵗ/ʷᵗ* and *ER-Cre⁺; mTmG⁺; Smad4ᶠˡ/ᶠˡ* mice, exposed to tamoxifen for 24 hr and incubated for an additional five days as described above. They were then incubated in Ca-Mg-free PBS and drawn in and out of a mouth-controlled micropipette to generate a suspension of denuded oocytes surrounded by the zona pellucida and individual or small clumps of granulosa cells. Briefly, the granulosa cells were deposited together with oocytes from CD-1 mice into the base of a 200 µl microfuge tube and PBS was added to a total volume of 100 µl. The tube was spun at top speed in a microcentrifuge for 1 min, then rotated 180° and centrifuged for 1 min. This step was repeated twice more. The tube was then cut near the base and the cell pellet was carefully scooped out and incubated for two days in four-well plates in MEM supplemented with FSH,

estradiol, and ITS. At the end of the incubation period, complexes of oocytes surrounded by granulosa cells were fixed and processed for analysis.

## Image analysis

To quantify the number of F-actin-stained-TZPs, a confocal optical section was obtained at the equatorial plane of the oocyte. Using Fiji software (National Institutes of Health, Bethesda, MD), a segmented circle was drawn around the oocyte in the middle of the zona pellucida, and the fluorescence intensity at each point on the line was obtained. Each point whose value was above the background value of the oocyte cytoplasm and higher than each of its immediately neighboring points was counted as a TZP.

To quantify EdU incorporation in granulosa cells, Fiji was used to calculate the ratio of Edu-labelled cells per GOC by dividing the number of EdU-positive cells by the total number of cells counterstained with DAPI.

To individually segment granulosa cells and their corresponding TZPs, Imaris 9.7.2 Software (Bitplane) was used. First, z-stacks of 0.2 µm thickness covering the middle region of a follicle were imaged to obtain a conventional 2D Maximum Intensity Projection (MIP). Then, a 3D rendered segmentation of GFP-positive granulosa cells was created with the 'Cell detection' tool where cell boundaries were detected based on GFP membrane staining. Smallest diameter was set to 100 µm, membrane detail of 0.2 µm and the threshold was adjusted based on local contrast. Segmented cells with a volume less than 100 µm$^2$ were deleted and fusing or fragmented cells were edited using the 'merge' or 'split' tools to match the original image. The oocyte was also segmented using the phalloidin red channel. After the edition, cells were converted to surface objects with the tool 'Convert to surfaces' to record their measurements. To render individual TZPs, the 'Filament tracer' tool and semi-automated tracing with the tool 'Autopath' was used to draw the path of each TZP from the granulosa cell body (start) to the oocyte surface (end) (*Figure 4—figure supplement 1A*). TZP diameter (thickness) was calculated automatically with a minimum diameter of 0.1 µm. To determine the distance of the TZP tip to the oocyte surface, the segmented TZP was automatically divided into discrete spots using the extension 'Filament to Spots.' This allowed the calculation of the distance from each spot to the oocyte surface. All measurements were exported using the 'distance to oocyte surface' from each TZP spot. To determine the TZP orientation (away or towards the oocyte), the distance of the spot located at the TZP start was subtracted from the distance of the spot located at the TZP tip. A positive difference indicates that the TZP tip is closer to the oocyte and a negative difference indicates that the TZP tip is farther from the oocyte (*Figure 4—figure supplement 1B*).

## Statistical analysis

Data are presented as mean ± standard error of the mean and were analyzed and represented using GraphPad Prism 8.4.2 software. Statistical significance was evaluated using one- or two-sample two-tailed t-tests.

## Acknowledgements

We thank Drs. Shibo Feng and Min Fun (RI-MUHC Imaging Platform) and Marie-Hélène Lacombe, Hélène Pagé-Veillette, and Dr. Ekaterina Iourtchenko (RI-MUHC Immunophenotyping Platform) for their invaluable assistance. We acknowledge the core facility CELLIM supported by MEYS CR (LM2023050 Czech-BioImaging) for their generous support and assistance in obtaining image analysis data using Imaris, and thank our colleagues for their advice and discussion. Supported by grants to HJC. from the Canadian Institutes of Health Research and the Natural Sciences and Engineering Research Council. HJC. gratefully acknowledges the support of the Richard Cruess Chair in Reproductive Biology at McGill University.

## Additional information

### Funding

| Funder | Grant reference number | Author |
|---|---|---|
| Canadian Institutes of Health Research | PJT153122 | Hugh J Clarke |
| Natural Sciences and Engineering Research Council of Canada | RGPIN-402138 | Hugh J Clarke |

The funders had no role in study design, data collection and interpretation, or the decision to submit the work for publication.

### Author contributions

Sofia Granados-Aparici, Conceptualization, Data curation, Formal analysis, Validation, Investigation, Methodology, Writing – original draft, Project administration, Writing – review and editing; Qin Yang, Data curation, Validation, Investigation, Methodology; Hugh J Clarke, Conceptualization, Data curation, Formal analysis, Supervision, Funding acquisition, Validation, Writing – original draft, Project administration, Writing – review and editing

### Author ORCIDs

Sofia Granados-Aparici (ID) https://orcid.org/0000-0003-4448-3254
Hugh J Clarke (ID) http://orcid.org/0000-0003-2626-244X

### Ethics

All experiments were performed in compliance with the regulations and policies of the Canadian Council on Animal Care and were approved by the Animal Care Committee of the Research Institute of the McGill University Health Centre (RI-MUHC, protocol 7783).

Reviewer #1 (Public Review): https://doi.org/10.7554/eLife.91798.3.sa1
Reviewer #2 (Public Review): https://doi.org/10.7554/eLife.91798.3.sa2
Author response https://doi.org/10.7554/eLife.91798.3.sa3

## Additional files

### Supplementary files

• MDAR checklist

### Data availability

Original data describing TZP characteristics have been deposited at Dryad https://doi.org/10.5061/dryad.tht76hf6s.

The following dataset was generated:

| Author(s) | Year | Dataset title | Dataset URL | Database and Identifier |
|---|---|---|---|---|
| Granados-Aparici et al. | 2024 | Data from: SMAD4 promotes somatic-germline contact during murine oocyte growth | https://doi.org/10.5061/dryad.tht76hf6s | Dryad Digital Repository, 10.5061/dryad.tht76hf6s |

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
