## [Editor Report · eLife assessment]

This study reports an **important** mechanism through which the TGF-beta signaling pathway promotes contacts between oocytes and their surrounding somatic cells by regulating the number of transzonal projections (TZPs) in mice. **Convincing** data support the conclusions. The work will be of interest to biomedical researchers who work on ovarian biology and female fertility.

---

## [Referee Report · Reviewer #1 (Public Review)]

Granados-Aparici et al., investigate somatic-germline interactions in female mice. Mammalian oocytes are nurtured in multi-cellular ovarian follicles and communication with surrounding somatic cells is critical for oocyte development. This study focused on transzonal projections (TZP) extending from granulosa cells to the surface of oocytes and document the importance of SMAD4, a TGF- β mediator, in regulating the TZPs. They propose a model in which individual TZPs contact the surface of the oocyte and stably attaches if there is sufficient N-cadherin. In SMAD4-depleted cells, there is insufficient N-cadherin to stabilize the attachment. The TZP continues to elongate but eventually retracts. Their model is well supported by their experimental evidence and the manuscript is both well-formulated and written.

Comments on revised version:

The authors have addressed the issues raised in the original review.

---

## [Referee Report · Reviewer #2 (Public Review)]

Summary:

This study proposed a new mechanism by which TGF-beta signaling pathway promotes contacts between oocyte and the surrounding somatic cells in mouse, by regulating the numbers of transzonal projections (TZPs).

Strengths:

The conditional Smad4 knockout and three-dimensional observation of transzonal projections are solid and sufficiently support the major conclusions.

Comments on revised version:

The authors have adequately addressed the reviewers' questions and comments.

---

## [Author Response]

The following is the authors’ response to the original reviews.

**Reviewer #1 (Public Review):**
Granados-Aparici et al., investigate somatic-germline interactions in female mice. Mammalian oocytes are nurtured in multi-cellular ovarian follicles and communication with surrounding somatic cells is critical for oocyte development. This study focused on transzonal projections (TZP) extending from granulosa cells to the surface of oocytes and documented the importance of SMAD4, a TGF- β mediator, in regulating the TZPs. They propose a model in which individual TZPs contact the surface of the oocyte and stably attach if there is sufficient N-cadherin. In SMAD4-depleted cells, there is insufficient N-cadherin to stabilize the attachment. The TZP continues to elongate but eventually retracts. Their model is well supported by their experimental evidence and the manuscript is both well-formulated and written.
**Reviewer #2 (Public Review):**
Summary:This study proposed a new mechanism by which the TGF-beta signaling pathway promotes contacts between oocytes and the surrounding somatic cells in mice, by regulating the numbers of transzonal projections (TZPs).Strengths:The conditional Smad4 knockout and three-dimensional observation of transzonal projections are solid and sufficiently support the major conclusions.Weaknesses:The physiological significance of SMAD4-dependent formation of transzonal projection networks is not assessed in this study.

Previous studies have shown that physical contact and gap junctional communication with the granulosa cells is essential for normal oocyte development. A recent study has also shown that depleting Myo10 in granulosa cells reduces the number of TZPs and leads to abnormalities in oocyte and embryo development. Thus, the importance of TZPs is well-established. These findings, which were insufficiently brought out in the Introduction of the original manuscript, have now been made more clearly (Introduction, 2nd paragraph). We recognize that these reports do not directly test a role for SMAD4-dependent TZPs. Unfortunately, it is beyond our technical capacity to obtain embryos following meiotic maturation and fertilization of oocytes that have grown in vitro, which wold be necessary for us to fully test the physiological role of SMAD4-dependent TZPs.

**Recommendations for the authors:**

**Reviewer #1 (Recommendations for the authors):**
(1) The authors switch from Amhr2-cre to ER-cre to increase the number of GFP-positive granulosa cells in 12 d/o ovaries. To avoid disruption of FSH secretion by SMAD4, they use an in vitro model that requires 6 days in GEO culture (1 d tamoxifen + 5 d). Could it be that Amhr2-cre didn't work because most follicles would not have reached the atretic preantral stage in 12 d/o ovaries? Did the authors consider 6 days in vitro GEO culture to determine if Amhr2-cre would be efficient and avoid exposure to tamoxifen?

Please see below.

When is Amhr2 expressed?

Previous studies (Jorgez et al, 2004; Pangas et al, 2006) report that Amhr2 is expressed in growing follicles that have progressed beyond a single layer of granulosa cells (often defined as secondary and primary follicles, respectively). As shown in Fig. 1C, we did not observe evidence of widespread Cre activity in multilayer follicles. At least two factors may contribute why we observed relatively weak Cre activity. One possibility is that, on the genetic background our mice, Amhr2 is expressed relatively late during follicular growth. Thus, we might have observed more GFP-positive granulosa cells in antral or pre-ovulatory follicles. Because the granulosa cells of these late-stage follicles would already have produced many TZPs, the number of new TZPs generated in wild-type but not SMAD4-depleted cells after Amhr2 activation would be a relatively small proportion of the total population. This would make it more difficult to detect a reduction in TZP number in the absence of SMAD4.

A second point is that we used pre-puberal mice whereas Jorgez et al examined Amhr2 expression in ovaries of adult mice. Pangas et al evaluated both prepuberal and adult females. It may be that Amhr2 is expressed earlier or more strongly in granulosa cells of adult mice. Regarding the suggestion to culture complexes obtained from mice on the Amhr2-Cre background, as this might allow widespread expression of Cre without the need for tamoxifen, this is an excellent idea. If there is considerable heterogeneity among cells in the timing of Amhr2-Cre activity, though, this may further cloud efforts to uncover the role of SMAD4 in the production or stability of TZPs, as noted above.

(2) Did most of the GEO cultured in vitro reach the antral follicle stage after 6 days?

Since GOCs were treated with collagenase, the thecal layer was removed. Therefore, development of an antrum does not occur. We observed that, in some cases, the oocyte was extruded from the granulosa cell mass. These abnormal complexes were discarded.

(3). Was the development/diameter of the oocyte in the GEO comparable to the oocyte growing in vivo?

We did not compare the diameter of the oocytes grown in vitro to those grown in vivo. Thus, we cannot say whether the oocytes grown in vitro reached the same size as those grown in vivo. We did, however, compare the diameter of the oocytes in the wt and ko groups and observed no difference (Figure 2). This indicates that depletion of SMAD4 in the granulosa cells does not impair oocyte growth. Importantly for our studies, it excludes the possibility that the reduction in TZP-number is simply due to a smaller surface area of the oocyte.

(4) SMAD4 depletion in granulosa cells disrupts steroidogenesis leading to increased progesterone levels and precocious luteinization of granulosa cells (Pangas et al., 2006). Did the authors determine the expression level of luteal markers of granulosa cells in the in vitro GEO culture Smad4 knockout model? Are their observations direct effects of the absence of SMAD4?

This is an excellent point. We checked our previously performed RNA-seq analysis of the wild-type and knockout granulosa cells, but found no difference in the quantities of Cyp11a1, Sfrp4, Star or Ptgfr. This is now described in the Discussion (4th paragraph). One potentially important difference between our study and that of Pangas et al (2006) is that they observed premature luteinization when prepuberal (3-week old) mice were injected with the FSH analogue, equine serum gonadotropin, whereas we studied granulosa-oocyte complexes cultured in vitro. This could underlie the apparent differences with respect to luteinization.

(5) Could the reduced number of TZPs in ER-cre+; Smad4fl/fl GOCs be explained by luteinization?

This interesting and logical possibility is related to the previous point. In other words, luteinization could be considered as a default pathway of differentiation that is suppressed by SMAD signaling. It is possible that luteinized cells are unable to generate or maintain TZPs. This model offers a potential mechanistic basis for our observation, and we now raise it in the Discussion (3rd paragraph).

**Reviewer #3 (Recommendations For The Authors):**
The expression and localization of N-cadherin should be observed in Smad4 and control granulosa cell-oocyte complexes.

We agree that this would be an excellent approach to confirm the decreased expression of N-cadherin in the granulosa cells that was observed by immunoblotting. We were confronted by two challenges, however. First, we were unable to consistently obtain strong staining of granulosa cell membranes in the inner layers of multilayer granulosa-oocyte complexes. Other antibodies are able to stain structures at the oocyte surface, indicating that antibodies are not physically blocked from penetrating the complex. More likely, the anti-N-cadherin does not bind its target strongly enough to generate a robust signal that can be detected through multiple overlying layers of cells. Second, whereas for immunoblotting we collect all granulosa cells from culture complexes, for immunofluorescence we are only able to examine those that remain in the complex. This means that, for immunofluorescence, we essentially but unavoidably select against cells that are only loosely attached – as would be expected for N-cadherin-deficient cells – to their neighbours. Given these challenges, we believe that the immunoblotting approach, which produced highly reproducible results over six biological replicates (Fig. 6), is the most reliable.